## Deriving hourly diagnostic surface velocity fields considering inertia and an application in the Yellow Sea

Sung-Won Cho<sup>1,2</sup>, Jang-Geun Choi<sup>3</sup>, Deoksu Kim<sup>4</sup>, Wenfang Lu<sup>5,6</sup> and Young-Heon Jo<sup>1\*</sup>

Correspondence to: Young-Heon Jo (joyoung@pusan.ac.kr)

**Abstract.** Surface currents play an important role in the transport of floating materials in the Yellow Sea, a region strongly influenced by tidal forcing and seasonal wind variability driven by the East Asian monsoon. While diagnostic models have been widely used to estimate surface currents, due to their steady-state assumption, high frequency variations such as tides and inertial oscillations cannot be resolved. To address this limitation, a time-dependent diagnostic model incorporating inertial terms into the governing equations is proposed. The performance of the proposed method is evaluated using buoy and drifter observations from 2015 to 2023. The time-dependent model in this study captures not only low frequency components (geostrophic and Ekman currents) but also high frequency variability (inertial oscillations and tides). Compared to the traditional model assuming steady-state, it shows significant improvement, achieving a correlation of 0.76 and Root-Mean-Square Error of 0.18 m s<sup>-1</sup> (compared to -0.08 and 0.43 m s<sup>-1</sup> for the steady model, that caused by wrong governing equation ignoring inertia to describe tides) because of successful consideration of high frequency variability. The decay rate of inertial oscillations is analytically derived, providing insight into the time scale for past signals in surface currents to dissipate. We expect that this study offers a practical framework for surface current estimation considering both high and low frequency signals and can be applied for quick assessments of material transport in other coastal oceans.

## 1 Introduction

Surface currents in oceanic environments are essential to understanding the transport of momentum and surface-floating materials. In the marginal sea such as the Yellow Sea (Figure 1), there are many issues caused by floating materials such as jellyfish blooms (Wei et al., 2015; Choi et al., 2018), harmful algal blooms (Onitsuka et al., 2010; Kim et al., 2016), green algae blooms (Son et al., 2015; Kwon et al., 2019), and Sargassum (Shin et al., 2022; Choi et al., 2023). Thus, there is an increasing demand for estimating surface velocity fields. Numerical hydrodynamic models are widely utilized to generate

<sup>&</sup>lt;sup>1</sup>Department of Oceanography and Marine Research Institute, Pusan National University, Busan, 46241, Republic of Korea.

<sup>&</sup>lt;sup>2</sup>Ocean Circulation and Climate Research Department, Ocean Circulation Research Center, Korea Institute of Ocean Science and Technology, Busan, 49111, Republic of Korea

<sup>&</sup>lt;sup>3</sup>Center for Ocean Engineering, University of New Hampshire, Durham, New Hampshire, United States.

<sup>&</sup>lt;sup>4</sup>Coastal Disaster & Safety Research Department, Korea Institute of Ocean Science & Technology, Busan, Republic of Korea.

<sup>&</sup>lt;sup>5</sup>School of Marine Sciences, Sun Yat-Sen University, Zhuhai, 519000, China

<sup>0 &</sup>lt;sup>6</sup>Southern Marine Science and Engineering Guangdong Laboratory (Zhuhai), Zhuhai, 519000, China

velocity fields and simulate the transport of surface-floating materials (Onitsuka et al., 2010; Son et al., 2015; Wei et al., 2015; Kwon et al., 2019), with advancements in data assimilation significantly enhancing their accuracy (Storto et al., 2019). Along with numerical hydrodynamic models, the diagnostic approach, which estimates surface velocity fields from sea surface height and wind datasets, has become a popular method due to its efficiency and accuracy (Bonjean and Lagerloef, 2002; Rio et al., 2014; Dohan, 2017; Choi et al., 2023).

The velocity fields from the previous diagnostic approach assumed that surface currents are primarily driven by the Ekman and geostrophic balance, so steady-state assumption was used. For instance, the Ocean Surface Current Analyses Real-time (OSCAR) velocity fields, provided by the National Aeronautics and Space Administration (NASA), were based on the diagnostic approach considering geostrophic and wind-driven Ekman current components (Bonjean and Lagerloef, 2002). Similarly, the Global total surface Current (GlobCurrent) dataset was produced by combining satellite-based geostrophic current and a semi-empirical model for the Ekman current component, aiming to accurately represent surface current variability (Rio et al., 2014). Recently, diagnostic velocity fields based on a simple analytical solution have been utilized and verified in the Yellow Sea to simulate algae transports and it was showed that velocity fields from the diagnostic approach can have comparable (or even locally better) accuracy to those from reanalysis datasets using numerical models and data assimilation (Choi et al., 2023). The seasonal Ekman currents driven by the East Asian Monsoon and the altimetry-based geostrophic currents, directed northeastward without a notable mean current pattern (orange arrows in Figure 1), successfully simulated the transport of surface floating substances in the Yellow Sea (Choi et al., 2018 and 2023).

Consequently, these diagnostic velocity fields developed in previous studies assumed steady states and considered only two low frequency signals (Ekman and geostrophic current components), so have limitations in capturing high frequency signals. In the Yellow Sea, there are significant high frequency variability of surface current driven by the near-inertial oscillations and tides (Teague et al., 1998; Hu et al., 2023). To consider these high frequency signals of surface current, new approach resolving the inertia is required. Time-dependent Ekman theory was advanced by Elipot and Gille (2009), Wenegrat and McPhaden (2016), and Lilly and Elipot (2021). These studies introduced and discussed analytical solutions incorporating the inertial terms, extending the classical steady Ekman theory (Ekman, 1905; Welander, 1957). Elipot and Gille (2009), in particular, assessed transfer functions of the analytical solutions based on spectral analysis of the observations.

Choi et al. (2023) showed that a simple analytical solution from the steady Ekman theory (Ekman, 1905; Welander, 1957) successfully reconstructs low frequency surface current in the Yellow Sea, that motivates this study to use a simple analytical solution from the time-dependent Ekman theory. This study builds techniques using the time-dependent Ekman theory to generate velocity fields considering both high and low frequency signals (Section 2). We expand the solution to consider time-dependency in the pressure gradient terms to resolve tidal velocity components. Based on the analytical solution and transfer function approach, diagnostic surface current velocity fields considering inertial term, which resolve high frequency variability, were derived and verified by buoy and drifter observations in the Yellow Sea. The velocity fields based on the time-dependent Ekman theory are compared with those based on the steady Ekman theory used by Choi et al. (2023).

60

50

Figure 1: Geographic locations of the study area. Red triangle indicates location of the Southern Jeju Island buoy from the Korea Hydrographic and Oceanographic Agency (KHOA). Yellow solid lines indicate the trajectories of drifters from the National Institute of Fisheries Science (NIFS) and solid circles indicate the initial launching positions of the drifters. The orange arrow field shows the geostrophic current fields averaged over the study period (2013 – 2023).

## 2 Theory

75

### 2.1 Governing equation and the analytical solution for time-dependent Ekman theory

The governing equation and vertical boundary conditions for time-dependent Ekman theory are given by

$$\frac{\partial \vec{u}}{\partial t} + f i \vec{u} = -g \frac{\partial \eta}{\partial \vec{n}} + A_z \frac{\partial^2 \vec{u}}{\partial z^2}, \qquad (1a)$$

$$A_z \frac{\partial \vec{u}}{\partial z} \Big|_{z=0} = \frac{\vec{\tau}^s}{\rho_0}, \qquad (1b)$$

$$\vec{u}|_{z=-h} = 0 \qquad (1c)$$

where  $\vec{u} = u + iv$  represents horizontal velocities in the complex coordinates, with i as the imaginary unit  $(i = \sqrt{-1})$ , t is time, f is the Coriolis parameter, g is gravitational acceleration,  $A_z$  is the vertical eddy viscosity assumed to be constant, and  $\partial \eta/\partial \vec{n} = \partial \eta/\partial x + i\partial \eta/\partial y$  is barotropic pressure gradient in the x and y direction and presumed to be a given complex time series. In the surface and bottom boundary conditions (1b) and (1c),  $\vec{\tau}^s = \tau_x^s + i\tau_y^s$  is the surface wind stress,  $\rho_0$  is a constant density, h is water depth. The governing equation (1) is identical to that of Choi et al. (2023) except for considering the inertial term. It must be noted that assuming constant eddy viscosity is a strong idealization, which can be relaxed through more complex analytical solution considering spatiotemporal variability of eddy viscosity (Elipot and Gille, 2009; Wenegrat and

McPhaden, 2016; Lilly and Elipot, 2021). Nonetheless, this assumption is retained here to focus on the impact of incorporating inertial terms and to ensure consistency with the formulation in Choi et al. (2023). Using simplicity of (1), the equation will be used to inversely estimate monthly representative  $A_z$  from the velocity observations (Section 3.2.2). By using constant eddy viscosity consistent with Choi et al. (2023), where steady Ekman theory was used, impact of considering inertial terms can be solely assessed by the comparison between steady and time-dependent (non-steady) Ekman theory (discussed in Section 4).

The boundary value problem (1) is given by a linear nonhomogeneous partial differential equation. It includes an interior forcing term given by the barotropic pressure gradient  $\partial \eta/\partial \vec{n}$  and surface wind stress forcing  $\vec{\tau}^s$  via surface boundary condition (1b). This partial differential equation can be solved using typical approaches using the superposition principle. Based on this principle, the total velocity can be decomposed as  $\vec{u} = \vec{u}_e + \vec{u}_g$ , where  $\vec{u}_e$  represents the wind-driven (Ekman) component, and  $\vec{u}_g$  denotes pressure-driven (geostrophic) component. The governing equations (1) can be separated into equations for each component:

95 
$$\frac{\partial \vec{u}_e}{\partial t} + f i \vec{u}_e = A_z \frac{\partial^2 \vec{u}_e}{\partial z^2} \qquad (2a)$$

$$A_z \frac{\partial \vec{u}_e}{\partial z} \Big|_{z=0} = \frac{\vec{\tau}^s}{\rho_0} \qquad (2b)$$

$$\vec{u}_e|_{z=-h} = 0 \qquad (2c)$$

100

$$\frac{\partial \vec{u}_g}{\partial t} + f i \vec{u}_g = -g \frac{\partial \eta}{\partial \vec{n}} + A_z \frac{\partial^2 \vec{u}_g}{\partial z^2}$$
(3a)  
$$A_z \frac{\partial \vec{u}_g}{\partial z} \Big|_{z=0} = 0$$
(3b)  
$$\vec{u}_g \Big|_{z=-h} = 0 .$$
(3c)

Note that (2) was solved and analyzed by Elipot and Gille (2009). For the arbitrary wind stress that can be expressed as Fourier series  $\vec{\tau}^s(t) = \sum_{n=-\infty}^{\infty} \hat{\tau}_n e^{-iw_n t}$ , the solution at z=0 is given by

$$\vec{u}_e = \sum_{n=-\infty}^{\infty} \frac{2}{\rho_0 D_{e_n} (f - w_n) j} \tanh\left(\frac{jh}{D_{e_n}}\right) \hat{\tau}_n e^{-iw_n t}$$
(4)

where j=1+i and  $D_{e_n}=\sqrt{2A_z/(f-w_n)}$  indicating the Ekman depth modified by the presence of the inertial term.  $H_{e_n}$  represents the transfer function discussed by Elipot and Gille (2009). Similarly, solution to (3) can be obtained by the typical approach to solve the linear nonhomogeneous differential equation (e.g., method of undetermined coefficients). With  $\partial \eta/\partial \vec{n}=\sum_{n=-\infty}^{\infty}\hat{\eta}_n'e^{-iw_nt}$ , the solution at z=0 is given by

$$\vec{u}_g = \sum_{n=-\infty}^{\infty} \underbrace{\frac{ig}{f - w_n} \left( 1 - \frac{1}{\cosh(jh/D_{e_n})} \right)}_{H_{g_n}} \hat{\eta}'_n e^{-iw_n t}$$
 (5)

where  $H_{g_n}$  represents the transfer function for the velocity component. The first and second terms in (5) represent the pressure gradient-driven (geostrophic) current and its bottom frictional response for the current. The detailed method calculating the velocity components using (4) and (5) will be illustrated in Section 3.2.

## 2.2 Dynamics of time-dependent Ekman theory and limitation of the classic steady Ekman theory

The solution to (1), given by the summation of (4) and (5), aligns with the steady generalized Ekman solutions (Ekman, 1905; Welander, 1957; revisited by Choi et al., 2023), extending them to include time-dependency of the momentum (inertia term). The steady state solution is given by

$$\vec{u}^* = \frac{2\vec{\tau}}{\underbrace{\rho_0 f D_e^* j}} \tanh\left(\frac{jh}{D_e^*}\right) + \underbrace{i\frac{g}{f} \frac{\partial \eta}{\partial \vec{n}} \left(1 - \frac{1}{\cosh(jh/D_e^*)}\right)}_{\vec{u}_g^*} \tag{6}$$

where D<sub>e</sub>\* = √2A<sub>z</sub>/f is the traditional (steady) Ekman depth. In this steady-state solution, the first term represents the Ekman component (\$\vec{u}\_e^\*\$), while the second term describes the geostrophic current and its bottom Ekman component (\$\vec{u}\_g^\*\$). The solution of time-dependent Ekman theory, (4) and (5), are almost identical to that of steady Ekman theory, (6), except that f in (6) is replaced by f - w<sub>n</sub>. Dynamics of the time-dependent Ekman theory without the pressure gradient term, (4), was discussed in the study by Lilly and Elipot (2021). The time-dependent Ekman depth D<sub>e<sub>n</sub></sub> becomes a function of the forcing frequency w<sub>n</sub>.
When w<sub>n</sub> « f (low frequency forcing), the dynamics are dominated by Coriolis force, governing classic steady Ekman and geostrophic currents. Conversely, when w<sub>n</sub> » f (high frequency forcing), Coriolis force becomes negligible relative to the inertia, and the system responds primarily with acceleration in the direction of the forcing. This response cannot be described by the classic steady Ekman theory, but is represented in the time-dependent solution, which better explains surface current responses under both low and high forcing frequency.

It is worth noting that resonance occurs when the frequency approaches the Coriolis frequency. In the limit of  $w_n \rightarrow f$  (resonance condition), (4) and (5) converge, respectively, to


$$\vec{u}_e = \frac{\hat{\tau}_n}{\rho_0} \frac{h}{A_z} e^{-ift}$$
 (7)

$$\vec{u}_g = -\hat{\eta}'_n \frac{gh^2}{2A_z} e^{-ift} \qquad (8)$$

which are the exact solutions to (2) and (3) in case of  $w_n = f$ , respectively. Consequently, when  $w_n \to f$ , the velocity amplifies to specific constants rather than  $\vec{u} \to \infty$  because of finite depth and the frictional bottom boundary condition.

## 3 Data and methods

## 3.1 Data







In this study, the velocity fields were derived using altimetric sea surface height and the corresponding geostrophic current fields provided by the Copernicus Marine Environment Monitoring Service (CMEMS), as well as wind stress fields from the ERA5 reanalysis dataset provided by the European Centre for Medium-Range Weather Forecasts (ECMWF). This configuration is identical to that of Choi et al. (2023) and Globcurrent velocity fields (Rio et al., 2014). The altimetric sea surface height and geostrophic current fields were provided at a spatial resolution of  $0.125^{\circ} \times 0.125^{\circ}$  and a daily temporal resolution. The altimetry measurements were processed using low-pass filtering to remove high frequency components, such as tidal and inertial oscillations, and then gridded using an optimal interpolation technique (Saraceno et al., 2008; Pujol et al., 2016; Taburet et al., 2019). The wind stress fields were provided at a spatial resolution of  $0.25^{\circ} \times 0.25^{\circ}$  and an hourly temporal resolution. Park et al. (2018) demonstrated that wind fields from the ECMWF reanalysis dataset show the highest comparability with in-situ observations around the Korean Peninsula. Ocean bathymetry, with a spatial resolution of 15 arc-seconds, was obtained from the General Bathymetric Chart of the Oceans (GEBCO) and is used as input for the steady and time-dependent Ekman theory models. TOPEX/Poseidon global tidal model (TPXO9) is a high-resolution global tidal model from Oregon State University used to estimate tidal elevation. TPXO9 has a horizontal resolution of  $0.033^{\circ} \times 0.033^{\circ}$ . The TPXO9 is a sophisticated inverse model from Oregon State Tidal Inversion Software (OTIS) based on numerous altimetric and tide-gauge measurements (Egbert and Erofeeva, 2002).

In addition, we validate the velocity fields using in-situ velocity measurements from buoy (Eulerian) and surface drifter (Lagrangian) measurements. Buoy data provided by the Korea Hydrographic and Oceanographic Agency (KHOA) (available at http://www.khoa.go.kr/oceangrid), which offers real-time and predictive information on oceanographic parameters (red triangle in Figure 1). The KHOA buoy dataset covers 2012 to the present; however, due to missing and erroneous data before 2015, the observations from 2015 to 2023 were used in this study. Outliers were removed by excluding values beyond five standard deviations. Drifter data were obtained from the Jellyfish Information System (yellow lines and circles in Figure 1). The drifter observations were conducted from 2015 to 2017 during summer seasons. We derived drifter velocities from the position changes by computing successive displacements at each timestep.

### 3.2 Methods

### 3.2.1 Surface current reconstruction

This study aims to estimate surface currents with high temporal resolution (hourly). To address this objective, the reconstruction process was designed to align the grids of the ERA5 wind stress and altimetric sea surface height fields. Therefore, ERA5 wind stress was spatially interpolated onto the grid of the altimetric sea surface height ( $\eta_{\text{altimetry}}$ ), and  $\eta_{\text{altimetry}}$  was then temporally interpolated using the Fourier transform interpolation, which conserves the original spectrum

of  $\eta_{\text{altimetry}}$ . It is worth noting that spiky errors are caused when linear interpolation is used in time (not shown here), so it is particularly meaningful to conserve the original spectrum of the data during temporal interpolation. For the pressure gradient forcing, as mentioned in Section 3.1, high frequency signals were removed in the  $\eta_{\text{altimetry}}$  during the gridding processes. To alleviate the absence of high frequency components in  $\eta_{\text{altimetry}}$ , sea surface height was reconstructed by simple summation of  $\eta_{\text{altimetry}}$  and tidal elevation  $\eta_{\text{tide}}$  based on TPXO9, so  $\eta = \eta_{\text{altimetry}} + \eta_{\text{tide}}$ . This is an ad hoc approach caused by lack of spatiotemporal sampling ability of observations. It must be noted that the reconstructed  $\eta$  incorporates both low frequency and several high frequency components (tidal harmonics) but cannot consider entire spectrum of real sea surface height (will be discussed in Section 5.3).

The surface currents are estimated based on (4) and (5) following the below steps:  $\hat{\tau}_n$  and  $\hat{\eta}'_n$ , depending on the frequency  $w_n$ , are obtained by Fourier transforming time series of  $\vec{\tau}^s$  and  $\partial \eta/\partial \vec{n}$ . And then,  $H_{e_n}$  and  $H_{g_n}$  are multiplied by  $\hat{\tau}_n$  and  $\hat{\eta}'_n$ , respectively. Finally, inverse Fourier transforms were performed to  $H_{e_n}\hat{\tau}_n$  and  $H_{g_n}\hat{\eta}'_n$ , that derive the surface current components  $\vec{u}_e$  and  $\vec{u}_g$ , respectively, in the time domain.

## 3.2.2 Optimal eddy viscosity parameter estimation







It is worth noting that more generalized analytical solutions resolving variability in the eddy viscosity are known (Elipot and Gille, 2009; Wenegrat and McPhaden, 2016; and Lilly and Elipot, 2021). Nevertheless, we used the solution assuming constant eddy viscosity, (4) and (5), because this study does not aim to examine the variability of eddy viscosity in the region, but to study influence of the inertia that was not considered in previous diagnostic velocity fields (Bonjean and Lagerloef, 2002; Rio et al., 2014; Choi et al., 2023). As mentioned above (Section 2), constant eddy viscosity was assumed for simplicity and consistency with the previous study but it cannot consider the seasonal variation of the eddy viscosity in the Yellow Sea (Hwang et al., 2014) influenced by the East Asian monsoon. To alleviate this issue, it is assumed that the eddy viscosity is a piecewise constant function, so the eddy viscosity is defined to be same constant value within a month. The value of the month was inversely estimated through sensitivity experiments by optimizing the eddy viscosity in  $\vec{u}_e$  to minimize discrepancies with the buoy observations. The observation for  $\vec{u}_e$  is defined as observed velocities minus altimetric geostrophic and tidal current components  $\vec{u}_{tide}$  that was calculated by T\_TIDE algorithm (Pawlowicz et al., 2002).

The eddy viscosity was optimized within the tested range of  $0.001 - 0.2 \text{ m}^2 \text{ s}^{-1}$  with  $0.001 \text{ m}^2 \text{ s}^{-1}$  interval. For each month, the analytical solution was computed using the predefined eddy viscosity. The Root-Mean-Square Error (RMSE) was calculated to measure the difference between the observed and estimated velocities and was then used as the objective function for optimizing the best-fit eddy viscosity estimation. Median values of the eddy viscosity estimated within each month (Figure 3) are used to calculate velocity fields in this study.

### 4 Results



## 4.1 Estimation of eddy viscosity based on the optimization approach

Figure 2 presents the optimized vertical eddy viscosity ( $A_z$ ) values, revealing distinct seasonal variability. The results indicate higher  $A_z$  values during winter (December – February), with a median value of approximately 0.17 m<sup>2</sup> s<sup>-1</sup> and a maximum reaching 0.2 m<sup>2</sup> s<sup>-1</sup>. In contrast, summer (June – August) exhibits lower  $A_z$  values, with a median around 0.02 m<sup>2</sup> s<sup>-1</sup> and minimal fluctuations. This seasonal variation is closely linked to the East Asian monsoon system, where stronger winter north-westerly winds enhancing vertical mixing, rather than weaker southerly winds during summer (Hwang et al., 2014). The results align with previous parameterization studies (Csanady, 1981; Cushman-Roisin and Beckers, 2011), which suggest that Ekman depth—and consequently  $A_z$  is strongly modulated by wind stress. Note that Csanady (1981) and Cushman-Roisin and Beckers (2011) parameterized the Ekman depth ( $D_e^*$ ) as a function of wind stress, where eddy viscosity can be expressed as  $A_z = f D_e^{*2}/2$ . Additionally, Choi et al. (2023) showed that this parameterization effectively applies to the Yellow Sea, reinforcing the robustness of these estimates. The performance of this optimization approach was similar with Choi et al. (2023) using the Ekman depth parameterization, that will be shown and further assessed in Section 4.2 through comparisons with observational data.

Figure 2: Box plots for the optimal eddy viscosity parameter for each month, minimizing the Root-Mean-Square Error (RMSE) between surface velocities derived from (3) and buoy velocities. The whiskers (plus symbols) indicate variability outside the upper and lower quartiles, and the blue line in the blue box shows the median value, that were used as representative values to calculate surface current.


# 4.2 Verification of time-dependent Ekman theory model with observation and comparison with the steady Ekman theory model

The comparison between velocities estimated using the time-dependent Ekman theory model and observed velocities is presented in Figure 3. Figures 3a and b show the comparison with buoy-observed velocities, showing correlation coefficients (R) of 0.70 and 0.66 for zonal and meridional components, respectively, and corresponding RMSE values of 0.15 m s<sup>-1</sup> for both components. Figures 4c and d show the comparison results with drifter-observed velocities. In this case, the correlation coefficients were higher than those for the buoy observations, with 0.78 for the zonal component and 0.82 for the meridional component. The RMSE for each component are 0.23 m s<sup>-1</sup> and 0.24 m s<sup>-1</sup>, respectively. As shown in Figure 1, the buoy observations (red triangle) are located in deeper ocean than the drifter trajectories (yellow circles and lines) concentrated in the shallower central and western Yellow Sea. The buoy location can be characterized by a stronger influence of the Kuroshio current branch (stronger geostrophic current component; Liu and Gan, 2012) and a weaker tidal current component (Hwang et al., 2014). This difference in measurement locations suggests that different hydrodynamic processes influence each observation dataset. Despite these regional differences, the model performs consistently across both observation locations, indicating its capability to represent surface current variability in the Yellow Sea. It is worth noting that the correlation with drifter observations is notably higher than that with buoy observations (Figure 3), likely due to the stronger tidal components, which are periodic so relatively easier to estimate accurately using a reliable  $\eta_{\rm tide}$ .

Figure 3: Comparison between time-dependent Ekman velocities with observations. (a) and (b) show comparisons with buoy-observed velocities for the zonal and meridional components, respectively. (c) and (d) display comparisons with drifter-observed velocities for the zonal and meridional components. The color scale indicates data density in logarithmic scale.

The steady Ekman theory model ( $\vec{u}^*$ ) based on Choi et al. (2023) and the time-dependent Ekman theory model ( $\vec{u}$ ) proposed in this study were compared to assess their performance in reconstructing surface currents (Figure 4). The results indicate that  $\vec{u}$  much more accurately represents observed surface currents, particularly in capturing high frequency variability, whereas  $\vec{u}^*$  shows notable limitations in this regard. This is reflected in the correlation coefficient of 0.78 and RMSE of 0.18 m s<sup>-1</sup>, whereas the steady model shows a significantly lower correlation of -0.08 and a higher RMSE of 0.43 m s<sup>-1</sup> (Figures 4a and b). Uncorrelated signals between the steady model and observation are because high frequency signals are not properly resolved by the governing equation ignoring inertia (Figure 4c and d), discussed in Section 2.2. It is worth noting that the previous studies considered only low frequency signal (Bonjean and Lagerloef, 2002; Rio et al., 2014; Choi et al., 2023). These results indicate that the  $\vec{u}$  properly represents high frequency variability in surface current, while the steady model has notable limitations.

Despite limitation of  $\vec{u}^*$  in capturing high frequency variability,  $\vec{u}^*$  remains effective in reconstructing low frequency surface current. Figures 4c and d compares the daily-averaged velocity from each model ( $\vec{u}$  and  $\vec{u}^*$ ) and that from observation. The steady model  $\vec{u}^*$  shows a correlation of 0.53 and RMSE of 0.10 m s<sup>-1</sup>, while time-dependent model  $\vec{u}$  shows a correlation of 0.62 and RMSE of 0.09 m s<sup>-1</sup> that is slightly improved than steady model. This similarity is expected, as the time-dependent Ekman model (4) and (5) mathematically converges to the steady Ekman model (6) when  $w_n \ll f$ . This explains why the steady Ekman theory can be used to simulate transport of the surface floating disturbances in the Yellow Sea (Choi et al., 2018 and 2023), of which time scale is order of a month much longer than 1/f, so expected to be governed by the low frequency dynamics (typical steady Ekman dynamics and predominant geostrophic balance). Consequently, the ability of the time-dependent Ekman model to represent both low and high frequency components stands out as the most significant. While the steady model can resolve only low frequency variability. We expect that the approach in this study can be applicable to the regions like the Yellow Sea where stress effect is dominant.

Figure 4: Comparison between observation (buoy and drifters) and velocity fields based on the time-dependent (left panels) and steady theory model (right panels). The upper panels compare observation with (a) time-dependent and (b) steady Ekman model velocities in hourly temporal resolution. The lower panels show comparisons for daily-averaged observation and (c) time-dependent and (d) steady models. The color scale indicates data density in logarithmic scale. Circles and triangles indicate zonal and meridional velocity components, respectively.

## **5 Discussions**

270

275

280

### 265 5.1 Limitation of the steady model

This study demonstrates that the steady Ekman theory model fails to represent high frequency variability in the observed velocities, whereas the time-dependent theory model provides a closer representation of the observations in both high and low frequencies. Figure 5 presents variance ellipses showing spatial patterns of surface current variability in observations (black solid ellipses), the steady theory models ( $\vec{u}^*$ ; blue ellipses), and the time-dependent theory model ( $\vec{u}$ ; red dashed ellipses) within  $0.5^{\circ} \times 0.5^{\circ}$  bins. The colormap within the grid represents the number of observations. The variance ellipse represents current variability: its orientation indicates the dominant direction of variability, and the lengths of the major and minor axes correspond to the variances along those directions. These ellipses reveal the differences between observed and modeled surface current variability. The ellipses estimated from the  $\vec{u}^*$  deviate from the observed patterns in both the magnitude and directional variability. This deviation becomes more pronounced toward the western coast of Korean Peninsula, where the ellipses appear enlarged and misaligned. In contrast,  $\vec{u}$  produces ellipses that more closely match observed ellipses, both in size and orientation, suggesting its improved capability in representing current variability. It is worth noting that the drifter observations are predominantly collected during the summer, leading to a seasonal bias, but the buoy observations include data from all seasons. The observed variance ellipses exhibit patterns that closely resemble the tidal characteristics, particularly the M2 tidal component, as reported by Hwang et al. (2014). This consistency further reinforces the dominant influence of tides on current variability in the study area. This suggests that  $\vec{u}$  effectively simulates tidal currents from the reconstructed  $\eta$ , shown in Section 3.2.1. Importantly, this underscores the role of the inertial term in representing tidal currents, as it enables the model to better represent high frequency variations.

Figure 5: Spatial distribution of observed (black solid line) and modeled variance ellipses by the steady Ekman model  $(\vec{u}^*;$  blue lines) and the time-dependent Ekman model  $(\vec{u};$  red dashed lines). The color scale indicates the number of observations in each bin. All variance ellipses are scaled consistently to represent the velocity magnitudes and directions within the respective grids. The steady model fails to resolve proper current variability, on the other hand, the variabilities are successfully considered by the time-dependent model.

Figure 6 shows Taylor diagrams for both the zonal and meridional velocity components, comparing the contributions of each current component, derived from  $\vec{u}$  and  $\vec{u}^*$ , to observations. The diagram provides a statistical evaluation of model performance using three key metrics: correlation, Standard Deviation (STD), and Root-Mean-Square-Difference (RMSD). The black lined circle markers represent observed velocities from drifters and buoys. Red markers represent components derived from  $\vec{u}$ , while blue markers indicate those from  $\vec{u}^*$ . When the wind-driven components from each model are compared, the time-dependent model ( $\vec{u}_e$ ; red squares in Figure 7) shows a higher contribution than the steady models ( $\vec{u}_e^*$ ; blue squares in Figure 6) so time-dependent model shows slightly improved agreement with observations. This implies that incorporating the inertial term enhances the model ability to represent wind-driven currents, particularly in response to high frequency wind forcing. The contribution of the time-dependency in the wind-driven component can be locally significant (e.g., typhoon event), that will be discussed in Section 5.2. The most significant limitation of the steady model is uncorrelated tidal components from  $\vec{u}_{tide}^*$  (blue downward-pointing triangles), on the other hand, the time-dependent model successfully resolves the tidal components  $\vec{u}_{tide}$  (red downward-pointing triangles). This alludes that the poor performance in the hourly steady model (Figure 4b) is due to misrepresentation for tides. In Figure 6,  $\vec{u}_g$  (red upward-pointing triangles) is located closer to the observations with reasonable correlation coefficients (0.69 and 0.72 for the zonal and meridional components, respectively). In contrast,  $\vec{u}_g^*$  (blue upward-pointing triangles) is uncorrelated with observation (-0.18 and -0.09 for the zonal and meridional

components, respectively) due to the issues in tides. These results explain the limitation of the steady model ignoring inertia that plays a key role in tides.

Figure 6: Taylor diagram between observations (drifters and buoy, black lined circles), time-dependent Ekman components (red markers) and steady Ekman components (blue markers) for the zonal components (a) and meridional component (b).  $\vec{u}$  (red circles) represents the total velocity,  $\vec{u}_g$  (red upward-pointing triangles) indicates the pressure-gradient component,  $\vec{u}_{tide}$  (red downward-pointing triangles) indicates tidal component, and  $\vec{u}_e$  (red squares) indicates wind-driven component, all from the time-dependent Ekman model.  $\vec{u}^*$  (blue circles) represents the total velocity,  $\vec{u}_g^*$  (blue upward-pointing triangles) indicates the pressure-gradient component,  $\vec{u}_{tide}^*$  (blue downward-pointing triangles) indicates tidal component, and  $\vec{u}_e^*$  (blue squares) indicates wind-driven component, all from the steady Ekman model.

It is worth noting that the contribution of each surface current component, explaining variance of the observation, can be quantified by the squared values of the correlation in Figure 6. The pressure-gradient component (red upward-pointing triangles in Figure 6), especially, tidal component (red downward-pointing triangles in Figure 6), is the most dominant contributor to surface current variability, and other components have relatively minor role in explaining the variance of the observation. However, in the longer time scales than the periods of tidal components (e.g., daily averaged velocity fields; Figure 7), contribution of the tidal components (red downward-pointing triangles in Figure 7) dramatically decreases and those of the other components, such as geostrophic current ( $\vec{u}_{alt}$ ; yellow upward-pointing triangles in Figure 7) and wind-driven current components (red squares in Figure 7), become considerable. Note that the geostrophic current component  $\vec{u}_{alt}$  is defined by the geostrophic balance equation and estimated by only altimetric sea surface height.

This is not surprising because the tidal components are purely periodic sinusoidal functions, whose means over their periods are intrinsically zero. Consequently, contribution of the velocity components varies with the time scale. For currents associated with the transport of the floating substances across the Yellow Sea, with a time scale on the order of months, non-tidal low frequency current components (geostrophic and Ekman currents) play the dominant role (Figure 7), even though tides outperform the other components on the short time scale (Figure 6). Based on the correlation shown in Figure 7, Ekman current components contribute slightly more than geostrophic currents in the Yellow Sea, that is consistent with Choi et al. (2018).

Figure 7: Taylor diagram comparing observed daily-averaged surface velocities (drifters and buoy, black-lined circles) with time-dependent Ekman model components (red markers) for (a) zonal and (b) meridional.  $\vec{u}$  (red circles) represent the time-dependent Ekman total velocity.  $\vec{u}_e$  (red squares) indicates wind-driven component,  $\vec{u}_g$  (red upward-pointing triangles) indicates the pressure-gradient component,  $\vec{u}_{\text{tide}}$  (red downward-pointing triangles) indicates tidal component.  $\vec{u}_{\text{alt}}$  (yellow upward-pointing triangles) indicates the geostrophic current component derived from altimetry ( $\eta_{\text{altimetry}}$ ).

## 5.2 Damped inertial oscillation caused by a typhoon



It is shown that tidal currents are the most dominant component in the variation of surface currents, but other components contribute to reconstructing more realistic surface currents in the study area (Figure 6). In this section, cases where other components (wind- and friction-driven components including inertial oscillations) become locally significant are shown and discussed. Figure 8 presents the time series of surface current responses to Typhoon Chanthu in 2021 (KMA, 2021) at the buoy site, including wind stress (Figures 8a and b), observed velocities from the buoy (black solid lines), and diagnostic

velocities estimates derived from approaches with (time-dependent theory model  $\vec{u}$ ; red solid lines) and without (steady theory model  $\vec{u}^*$ ; blue solid lines) the inertial term.

The M2 tidal constituent is dominant (Figures 8c and d) and the time-dependent model properly captures the signals from M2 tidal component, on the other hand, steady model exhibits poor phase agreement with observation. In addition, considerable wind-driven component is also shown during the typhoon event (blue-shaded area in Figure 8). Between September 16 and 20, which includes the typhoon-affected period, non-tidal components became even more dominant than tidal component; correlation coefficients between observed (black solid line in Figure 8c and d) and reconstructed currents without tides (red solid line in Figure 8e and f) reached 0.73, which is notably higher than correlation of tidal component, 0.45, based on T\_TIDE. This indicates that, although tide generally dominates, the other (wind-driven) components play a significant role under strong wind events like typhoons.







Removing the tidal component from observation  $(\vec{u} - \vec{u}_{\text{tide}})$  reveals clear identification of typhoon-induced inertial oscillations after the typhoon event (Figures 8e and f). The steady model fails to represent the inertial oscillation, as it only reflects low frequency variations. This limitation is evident as the steady model neglecting inertia cannot consider the inertial oscillation, governed by balance between the Coriolis force and inertia. In contrast, the time-dependent model clearly resolves the inertial oscillation and also its damping (Figure 8e and f). Following the abrupt perturbation from the typhoon, inertial oscillations are developed as the current responds to the typhoon and then gradually decay over time, reflecting dissipative processes in post-storm currents (Figures 8e and f). Owing to the simplicity of the governing equation (1), the decay rate can be analytically determined by expanding boundary value problem (1) to initial-boundary value problem with initial condition  $\vec{u}(z,t=0) = \vec{u}_0(z)$  where  $\vec{u}_0(z)$  is arbitrary initial velocity profile. Detailed mathematical derivations are in the Appendix. Considering an arbitrary initial condition to (1) yields additional velocity component  $\vec{u}_i$  associated with the initial condition is given by

$$\vec{u}_i = \sum_{n=0}^{\infty} e^{-ift} e^{-\gamma_n t} \vec{U}_n \cos(k_n z)$$
 (9)

where  $\gamma_n = A_z k_n^2$  indicates the decay rate of *n*-th mode and  $\vec{u}_i(z,t=0) = \vec{u}_0(z) - \vec{u}_e(z,0) - \vec{u}_g(z,0) = \sum_{n=0}^{\infty} \vec{U}_n \cos(k_n z)$ . The inertial oscillations and attenuations are described by  $e^{-ift}$  and  $e^{-\gamma_n t}$  in (9), respectively. It is worth noting that the decay rate  $\gamma_n$  depends on the wavenumber  $k_n$  of vertical velocity profile. Therefore, it is difficult to determine precise decay rate of the surface current without information about the vertical velocity profile. Nevertheless, the slowest (minimum) decay rate occurs when n=0, so  $\gamma_0=A_z k_0^2=A_z(\pi/(2h))^2$ , which is not only applicable to surface current but also the most important value because its reciprocal indicates the longest time scale for the signals to arrive. In Figures 8e and f, the red dashed line indicates exponential decay using the slowest decay rate and  $A_z$  value for September (shown in Figure 2). It successfully explains decay of the inertial oscillation shown in the time-dependent theory model  $\vec{u}$ .

The time-dependent model shows slightly stronger damping compared to the observation (Figure 8e and f). Despite the overestimated damping, it must be highlighted that the time-dependent model successfully reconstructs the overall surface

currents in the Yellow Sea (Figure 3). The time-dependent model plausibly captures the damped inertial oscillation, that is totally ignored by the steady model (Figure 8e and f). The simplicity of theory model provides valuable insight into the attenuation of inertial oscillations. It can be explained how vertical eddy viscosity and bottom friction attenuates signals that occurred in the past: the decay rate depends on the wavenumber of vertical velocity profile but the slowest damping rate can be applicable to surface current. Ultimately, this study underscores the advancement of diagnostic approaches in high frequency processes with the inclusion of the inertial term.



Figure 8: Time series comparison of buoy-observed velocities (black lines) and modeled velocities from the time-dependent (red lines) and steady (blue lines) Ekman theory models during Typhoon Chanthu in September 2021. The

period when the typhoon passed over the buoy observation location is highlighted by blue shading. Upper panel (a) and (b) show time series of wind stress at the buoy site. Middle panel (c) and (d) show direct comparison between observed and modeled velocities. Lower panel (e) and (f) show comparison of de-tided velocity time series, where the red-dashed line represents the arbitrary exponential function using  $\gamma_0 = A_z(\pi/(2h))^2$ , with  $A_z$  corresponding to the September eddy viscosity shown in Figure 2.

## 5.3 Resolved and unresolved spectrums by the time-dependent Ekman theory model





Figure 9 presents the spectrum of zonal (upper panels) and meridional (lower panels) velocity components. The left panels (Figure 9a and c) show buoy-observed velocities while the right panels (Figure 9b and d) show reconstructed velocities from the time-dependent Ekman model. The red vertical dashed line indicates the local Coriolis frequency  $(f/2\pi)$ , and the labeled peaks correspond to dominant tidal constituents: the diurnal (O1 and K1), and semidiurnal (M2, S2 and N2) tidal components. To highlight the dominant spectral patterns, band-averaged spectrum (bold solid lines) is also shown.

The time-dependent model (right panels) clearly represents tidal components, showing distinct spectral peaks at specific constituents in observation (left panels). In addition to low-frequency signals, the model spectrum also represents the resonant response near the inertial oscillation frequency. As discussed by Kim and Crawford (2014) and Elipot and Lumpkin (2008), resonance occurs when the frequency of external forcing is close to the  $f/2\pi$ , amplifying the ocean's response to the forcing. This spectral peak near the  $f/2\pi$  is shown in the observation (black solid lines in left panels) and resolved by the model (red solid lines in right panels). Consequently, the time-dependent Ekman theory model  $\vec{u}$  accurately resolves the resonant signals near the  $f/2\pi$  although the model underestimates high frequency signals.

Figure 9: Spectrum amplitude of observed (left panels (a) and (c)) and modeled (right panels; (b) and (d)) velocity for the zonal (upper panels; (a) and (b)) and meridional (lower panels; (c) and (d)) components. Blue lines represent the raw spectra. The bold solid lines represent the band-averaged spectra with 100-point window from observations (bold black lines in left panels) and the time-dependent model (bold red lines in right panels), respectively. Major tidal constituents (O1, K1, M2, S2, and N2) are indicated by yellow markers, and the vertical dashed red line denotes the local Coriolis frequency  $(f/2\pi)$ .



This underestimation can be attributed to the uncertainties in the reconstruction of  $\eta$  using the ad hoc approach shown in Section 3.2.1. Since the pressure-gradient component is calculated by the reconstructed  $\eta$ , its accuracy directly impacts spectral characteristics. The spectrum of  $\partial \eta/\partial x$  (Figure 10a) and  $\partial \eta/\partial y$  (Figure 10b) reveals a limitation in the reconstruction approach, that considers only low frequency signals estimated from altimetry and several high frequency modes corresponding tidal harmonics. The simple summation of the low frequency component  $\eta_{\text{altimetry}}$  and the tidal component  $\eta_{\text{tide}}$  fails to fully capture high frequency variability and misses many intermediate and high frequency modes (Figure 10). Consequently, the ad hoc  $\eta$  reconstruction is not sufficient to resolve entire spectrum. We expect that this limitation can be alleviated as availability

for  $\eta$  observation increases, for example, by using blended sea surface height field by tide gauge data. Nonetheless, this study remains valuable as it demonstrates the ability to capture high frequency variability including tides and inertial oscillations, which are predominant in the study area.

Figure 10: Spectrum amplitude of the reconstructed sea surface height gradients: (a)  $\partial \eta/\partial x$  and (b)  $\partial \eta/\partial y$ . Blue lines indicate raw spectra and black bold lines represent band-averaged spectra using a 100-point window. Major tidal constituents (O1, K1, M2, S2, and N2) are marked with yellow triangles, and the vertical red dashed line indicates the local Coriolis frequency  $(f/2\pi)$ . These spectra show limitation of the simple sea surface height reconstruction that

resolves only low frequency signals from altimetry and several high frequency modes from tides. Many intermediate

10<sup>-10</sup>

### **6 Conclusion**



10<sup>-10</sup>

and higher frequency signals are not resolved.

The usefulness of the diagnostic surface velocity field was reported by many previous studies (Bonjean and Lagerloef, 2002; Rio et al., 2014; Choi et al., 2018; Choi et al., 2023), but steady models considering only geostrophic and Ekman current components have been used. To the best of our knowledge, this is one of the first studies developing the diagnostic velocity field considering the inertial terms in the momentum equations, along with Ubelmann et al. (2025). Although their response function approach, in time domain, is equivalent to our transfer function method, in frequency domain, based on the convolution theorem, there are key differences. To be specific, they used data-driven empirical modeling approaches for the response function, whereas we adopted a stricter formulation based on the analytical solution to the momentum equation. It is worth noting that Ubelmann et al. (2025) used a slab model, which is vertically averaged, for their rationale and discussion. On the other hand, the governing equations of this study are not vertically averaged. The fact that we generalized the time-dependent Ekman theory to incorporate pressure gradient term, thereby resolving tides, is our unique contribution.

This study evaluates the performance of the time-dependent Ekman theory model and compares the model with the steady Ekman theory model (explicitly and implicitly used by the previous studies) in the Yellow Sea. The time-dependent

model successfully considers not only low frequency velocity components (geostrophic and Ekman currents) but also high frequency components (inertial oscillations and tides). The tidal currents outperform the other velocity components, such as geostrophic and Ekman current components, at short time scales described by the hourly diagnostic velocity fields (Figure 6). On the other hand, the contribution of tides in longer time scales becomes negligible because averaged values over their periods are zero (Figure 7). This implies that the dominant current component transporting a subject is determined by the time scale: surface floating substances crossing the Yellow Sea, whose time scale is order of months, are dominantly determined by low frequency velocity components including geostrophic and Ekman current components. Once position of the subject should be forecasted as quickly as possible on short time scale (e.g., search and rescue), tides become considerably important. This study also highlights the role of the simple analytical solutions improving our understanding for the high frequency surface current variability and provides a pathway for future advancements in surface current estimation.

## **Appendix**






This appendix provides the mathematical formulation of the expanded initial-boundary value problem given by

$$\begin{split} \frac{\partial \vec{u}}{\partial t} + f i \vec{u} &= -g \frac{\partial \eta}{\partial \vec{n}} + A_z \frac{\partial^2 \vec{u}}{\partial z^2}, \qquad (A1a) \\ A_z \frac{\partial \vec{u}}{\partial z} \bigg|_{z=0} &= \frac{\vec{\tau}^s}{\rho_0}, \qquad (A1b) \\ \vec{u}|_{z=-h} &= 0 \qquad (A1c) \\ \vec{u}|_{t=0} &= \vec{u}_0(z). \qquad (A1d) \end{split}$$

It must be noted that the solution (4) and (5) discussed in Section 2 are from the boundary value problem (1), so there is no consideration of specific initial condition. The only difference from (1) is the inclusion of the initial condition (A1d), which defines an arbitrary initial velocity profile. The total velocity  $\vec{u}$  is decomposed as  $\vec{u} = \vec{u}_b + \vec{u}_i$ , where  $\vec{u}_b$  and  $\vec{u}_i$  are the component associated with forcings and initial condition, respectively. The governing equation and boundary condition for  $\vec{u}_b$  are identical to (1) so given by

$$\frac{\partial \vec{u}_b}{\partial t} + f i \vec{u}_b = -g \frac{\partial \eta}{\partial \vec{n}} + A_z \frac{\partial^2 \vec{u}_b}{\partial z^2}, \qquad (A2a)$$

$$A_z \frac{\partial \vec{u}_b}{\partial z} \Big|_{z=0} = \frac{\vec{\tau}^s}{\rho_0}, \qquad (A2b)$$

$$\vec{u}_b|_{z=-h} = 0 \qquad (A2c)$$

Therefore, solution to (A2) is the time-dependent theory model  $\vec{u}_b = \vec{u}_e + \vec{u}_g$  where  $\vec{u}_e$  and  $\vec{u}_g$  are given by (4) and (5). Subtracting (A1) and (A2) yields the governing equation and boundary conditions for  $\vec{u}_i$  given by

$$\frac{\partial \vec{u}_i}{\partial t} + f i \vec{u}_i = A_z \frac{\partial^2 \vec{u}_i}{\partial z^2}, \quad (A3a)$$

$$A_z \frac{\partial \vec{u}_i}{\partial z} \bigg|_{z=0} = 0, \qquad (A3b)$$

$$\vec{u}_i|_{z=-h} = 0. \tag{A3c}$$

The initial condition for  $\vec{u}_i$  can be obtained by rewriting (A1d) and given by

$$\vec{u}_i(z, t = 0) = \vec{u}_0(z) - \vec{u}_b(z, 0).$$
 (A4)

Consequently, (A3) and (A4) become a problem without forcing terms,  $\vec{\tau}^s/\rho_0$  and  $-g\partial\eta/\partial\vec{n}$ . Solution to (A3) with the initial condition (A4) is given by (9) and can be obtained by typical approach to solve the heat equation.

### 475 Data availability




Sea surface height data are freely available at <a href="https://doi.org/10.48670/moi-00148">https://doi.org/10.24381/cds.adbb2d47</a>. TPXO9-atlas barotropic tidal solution are provided by the Oregon State University (USA) and is available online for academic research and other noncommercial uses under previous registration (at <a href="https://www.tpxo.net/global/tpxo9-atlas">https://www.tpxo.net/global/tpxo9-atlas</a>). Water depth data were downloaded from the freely available GEBCO Compilation (2024) at <a href="https://download.gebco.net/">https://download.gebco.net/</a>. The surface current algorithm and drifter data utilized in this study will be made publicly available upon acceptance of this manuscript.

### **Author contribution:**

Cho, S.-W.: Conceptualization, Investigation, Formal analysis, Visualization, Writing - Original Draft. Choi, J.-G.: Methodology, Software, Visualization, Writing and provided mentorship to S.-W. Cho. Kim, D.: Methodology, Visualization, Validation. Lu, W.: Methodology, Visualization. Jo, Y.-H.: Writing - Review & Editing, Supervision.

### **Acknowledgments:**

This study was supported by Korea Institute of Marine Science & Technology (KIMST) Promotion funded by the Ministry of Oceans and Fisheries [(Grant No. 20210046, Development of technology using analysis of ocean satellite images) and (Grant No. RS-2023-00256330, Development of risk managing technology tackling ocean and fisheries crisis around Korean Peninsula by Kuroshio Current)].

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
