# Peer review of "Deriving hourly diagnostic surface velocity fields considering inertia and an application in the Yellow Sea"

_EGUsphere, 2025_

## Author Response (AR2)

**Reviewer 1**

This manuscript employs a diagnostic model to derive the surface velocity fields in the Yellow Sea. It presents many comparisons between the steady state Ekman model and the time-dependent Ekman model, showing that the time-dependent Ekman model has a great improvement. This seems to be quite simple since we must use the time-dependent one when considering the velocity of higher frequencies, such as tide and inertial. And the same feeling for including the inertial term, which is absolutely important at the period of strong wind change as it generates significant near-inertial currents. And including the time-dependent and inertial parts is easy, and not new. Overall, there are little novel insight in this work, and its scientific significance is low.

We appreciate the reviewer's comments and efforts for our study. As the reviewer mentioned, inertial term is essential for resolving high-frequency processes such as tides and near-inertial oscillations and previous studies have already addressed this importance of inertia term, that was acknowledged in our manuscript (Line 48 in original manuscript).

Nevertheless, it has not been considered in previous studies on diagnostic velocity fields (Bonjean and Lagerloef, 2002; Rio et al., 2014; Dohan, 2017; Choi et al., 2023; cited in the manuscript). It is worth noting that OSCAR and GlobCurrent (currently provided by the most famous agencies, NASA and CMEMS) are still based on steady-state formulations ignoring the inertial term. To the best of our knowledge, our study is the first study applying the time-dependent Ekman theory to the diagnostic surface current reconstruction, which represents the novel contribution of our work. We kindly argue that, even though the theory and dynamics are not new, the application testing its surface current reconstruction ability is new and scientifically significant.

Furthermore, we generalized the analytical solution used in the theory to account for the pressure gradient, which may be regarded as progress in theory, and it enables the diagnostic velocity field to resolve tidal component. The fact that method is easy is what we intended, so many other researchers can easily adopt the method proposed in this study for their study areas and own dataset.

Line 25: 'coastal oceans' is not propriate.

We have revised the wording to "marginal sea" to more accurately describe the Yellow Sea (L26).

Line 155: The time range of drifter data should be noted, as the comparison probably has a seasonal difference.

In the revised manuscript, the explicit time range of the drifter dataset have been added in the Data section (L163). We appreciate this valuable suggestion about the seasonal comparison. Unfortunately, as we mentioned in manuscript, drifter observations are not available in winter, so we examined buoy observation for both summer and winter (Figures. S1). There are not significant differences between summer and winter in terms of skill scores. The seasonality of the surface current system of the study area, governed by low-frequency Ekman-geostrophic balances, were well discussed by Choi et al. (2023). In this study, we would like to focus on high frequency dynamics.

Figure S1. Comparison of buoy observations with modeled velocities for different seasons and components. (a) and (b) show summer, while (c) and (d) show winter. Zonal and meridional components are displayed in the left and right panels, respectively. The logarithmic color scale represents data density.

Line 216: How to obtain the velocity from drifters should have more detail. The buoy movement is affected not only by the surface current, but also by the direct wind push through a drag coefficient and the Stoke drift induced by the surface wave. Do you consider them?

Thank you for this helpful comment. We obtained drifter velocities from the observed positions (latitude and longitude) by calculating successive positions over next timestep, that is added in the manuscript (L163).

The direct wind pushing (e.g., leeway drift) and Stoke drift are not considered in this study, similarly with the other diagnostic velocity field did (Bonjean and Lagerloef, 2002; Rio et al., 2014; Choi et al., 2023; cited in the manuscript). In this study, we focus on the merit of the diagnostic velocity field in considering the inertia term. We expect that the incorporating the velocity components (leeway and Stoke drifts) will enhance the diagnostic velocity field, but the results in this study (e.g., Fig. 3) elucidate that most variations in the in-situ observation can be explained by the four terms (inertial, Coriolis, pressure gradient, and vertical eddy viscosity).

Line 280: It is not clear what the variance ellipse stands for

Following the reviewer's comment, we have added a statement explaining the variance ellipse in the revised manuscript. The following sentences will be added in Section 5.1 of the revised manuscript (L276):

Variance ellipse represents current variability: the orientation indicates the dominant direction of variability, and the length of major and minor axis corresponds to the variances along those directions.

**Reviewer 2**

In this study, the authors implement a diagnostic model for surface flows in the Yellow Sea, using a time-dependent model that includes both low-frequency geostrophic and Ekman velocities, and high frequency flows such as tides and inertial oscillations. The model in this setting, appears to substantially improve on earlier work using a steady Ekman model.

The authors do a good job demonstrating the advantage of incorporating the inertial terms in the diagnostic velocity field and comparing it to the steady Ekman theory model.

We appreciate the reviewer's insightful comments and are grateful for the positive assessment. We address each point below and explain the revisions made to improve the manuscript.

I would recommend accepting the manuscript with minor revisions:

My main reservation is about the geostrophic component obtained from altimetry, and I believe the authors would improve the manuscript by attempting to address some of these concerns:

(1) What kind of geostrophic circulation features are recurring in the Yellow Sea? It may be mentioned in past studies, but it would help to add a figure or two in the introduction or in section 3.2.1, documenting mesoscale motions and their variability.

We thank the reviewer for this helpful suggestion. Following the reviewer's comment, we added statements reviewing the current pattern of the Yellow Sea discussed in Choi et al. (2018). In addition, temporal-mean geostrophic current fields are added in Figure 1. Because Choi et al. (2018) already showed and discussed the low-frequency velocity component fields of the Yellow Sea, including the geostrophic circulation features and its dominance relative to the Ekman current (e.g., Fig. 4 in Choi et al., 2018), we would like to avoid profound discussions about the low-frequency components to focus on the high-frequency signals, such as tides and near-inertial oscillations, that are main modeling subject of this study.

Figure 1 (revised). Climatological mean geostrophic velocity fields are added.

(2) What is the contribution of the geostrophic component for the surface velocities in the Yellow Sea, with respect to the tidal and Ekman velocities?

We strongly agree with the importance of the contribution of each velocity component. This is one of the reasons we provided the correlation (Fig. 6), whose squared value represents the variance of the observation explained by the component. We revised the manuscript (L313-327 in revised manuscript) to clarify relative contribution of each velocity components with additional discussion below.

In terms of the squared correlation  $(R^2)$  based on the hourly velocity fields, the variance explained by the tidal component  $(\vec{u}_{tide})$  is predominant  $(R^2=0.49)$  relative to that explained by geostrophic  $(\vec{u}_g)$  and Ekman  $(\vec{u}_e)$  current components  $(R^2=0.02 \text{ and } R^2=0.07)$ , respectively). However, when the velocity fields are daily averaged (Fig. 7 in revised manuscript; also shown below), this pattern reverses: the tidal current contribution (red downward-pointing triangles) becomes negligible  $(R^2=0.03 \text{ in Fig. 7})$ , whereas the geostrophic  $(\vec{u}_{alt})$ , yellow upward-pointing triangles) and Ekman components (red squares) becomes considerable  $(R^2=0.17 \text{ and } R^2=0.23)$ , respectively; Fig. 7). This attenuation of the tidal signal is not surprising, since the tidal currents are purely periodic (sinusoidal functions) and their temporal means over the tidal periods are intrinsically zero.

Consequently, if research interests are the floating materials crossing the Yellow Sea (Choi et al., 2018 and 2023), their time scales are order of a month, much longer than periods of tides, so

geostrophic and Ekman currents become more dominant than tides (mentioned in L250-254). On the other hand, if we need to forecast positions of subjects as quickly as possible on short time scales (e.g., search and rescue; SAR), tides become considerably important. Those contents are added in discussions (L313-327) and conclusion (L434-441) sections.

Figure 7. Taylor diagram comparing observed daily-averaged surface velocities (drifters and buoy, black-lined circles) with time-dependent Ekman model components (red markers) for (a) zonal and (b) meridional.  $\vec{u}$  (red circles) represent the time-dependent Ekman total velocity.  $\vec{u}_e$  (red squares) indicates wind-driven component,  $\vec{u}_g$  (red upward-pointing triangles) indicates the pressure-gradient component,  $\vec{u}_{\rm tide}$  (red downward-pointing triangles) indicates tidal component.  $\vec{u}_{\rm alt}$  (yellow upward-pointing triangles) indicates the geostrophic current component derived from altimetry ( $\eta_{\rm altimetry}$ ).

(3) More specifically regarding the altimetry product: how close is the altimetry-derived geostrophic component to the in-situ velocities? Are the (CMEMS) velocity field errors (due to spatiotemporal interpolations, and quite shallow depth of the Yellow Sea) distinct from the high frequency motion errors, which are the main focus of this study? For instance, if you were to remove the geostrophic component in the diagnostic model, how much would it affect the correlations and RMS errors?

We agree with the reviewer's comment and expect that it is one of the significant factors influencing performance of the diagnostic velocity fields in the Yellow Sea. However, our

previous studies (Choi et al., 2018 and 2023) showed that the geostrophic velocity fields work reasonably. Furthermore, this study also showed that the diagnostic velocity fields using altimetry-based geostrophic currents successfully resolve low frequency variations of the observations (Fig. 4c and d). This implies that the altimetry-based geostrophic velocity fields are good enough to resolve the true geostrophic velocity fields.

We have checked the standard error provided by CMEMS geostrophic current product. The error reported in the Yellow Sea is less than  $0.05 \text{ m s}^{-1}$ , which is much smaller than the errors we estimated ( $\sim 0.2 \text{ m s}^{-1}$ ; Figs. 3 and 4). Removing the geostrophic component has little effect at the hourly scale (R slightly decrease from 0.76 to 0.74 and RMSE increase from 0.18 to 0.19). In the daily averaged velocity field, when it is removed, R decrease from 0.62 to 0.51 and RMSE increases from 0.09 to  $0.10 \text{ m s}^{-1}$ . This shows that the altimetry-based geostrophic currents are improving performance of the model.

**- Typos:**

We thank the reviewer for pointing out these typographical and grammatical errors. We have corrected them accordingly in the revised manuscript

line 43: "(Choi et al., 2023). Choi et al. (2023)"

The statement is modified in the revised manuscript for better readability (L43).

line 64: "trajectories of drifter"

We appreciate the reviewer's comment. The phrase "trajectories of drifter" has been corrected to "trajectories of drifters" for grammatical accuracy (L66).

line 179: "..which, as a result,.."?

Thank you for the suggestion. The sentence has been revised for smoother connection and improved readability (L183).

line 248-249 (rephrase?) " This explains that the reason the steady Ekman theory.." to "this explains why the steady Ekman theory.."

We agree with the reviewer's suggestion. The phrase is revised following the reviewer's comments (L249).

line 393: "not sufficiently" -> "not sufficient"

Thank you for pointing this out. The phrase "not sufficiently" has been corrected to "not sufficient" (L412).

**Reviewer 3 (Editor)**

1 – There is a manuscript that seems to have a similar aim, that was submitted on march 2025 and is now accepted (Ubelmann et al.)

https://egusphere.copernicus.org/preprints/2025/egusphere-2025-1149/. I suppose that it is relevant for you to cite this manuscript and explain how your work differs or agrees with it.

We appreciate the editor's suggestion and have cited Ubelmann et al. (2025) in the revised manuscript. Differences between Ubelmann et al. (2025) and this study are stated in L430-436 (empirical modeling vs theoretical analytical modeling; vertically averaged slab model vs governing equations without vertical averaging; absence and presence of pressure gradient term). We would like to highlight that generalization of the time-dependent Ekman theory to incorporate the pressure-gradient term is our unique contribution.

2 – Why do you derive the tidal velocities from the sea surface height of TPX09 rather than using the tidal velocities u, v provided by TPX09, which I presume would be much more accurate? It would be fair at least to compare your tidal velocities with TPX09 and comment on your choice of method.

We used the tidal elevation from TPXO9 to derive sea surface height fields resolving both highand low-frequency components. Although TPXO9 provides very accurate velocity components for specific tidal constituents, it can resolve only tidal components. On the other hand, our approach allows representation of all frequency bands beyond the tidal constituents (e.g., successfully resolved low frequency components of pressure gradient-driven currents corresponding to the geostrophic currents).

This approach also demonstrates better extensibility. For example, many other tidal inverse models (e.g., FES2014 and EOT20) provide only tidal elevations without velocity fields, and thus our method can be used to estimate velocity component from these tidal models. As we mentioned in L414-417, once better sea surface height fields, resolving intermediate spectrums between tides (from inverse tide model) and geophysical scale variations (captured by altimetry), become available, we expect that our method which can resolve non-tidal components should be adopted.